# Exploring the factors behind socioeconomic inequalities in Antenatal Care (ANC) utilization across five South Asian natiaons: A decomposition approach

**Mortuja Mahamud Tohan**⬤*, **Md. Amirul Islam, Md. Ashfikur Rahman**

Development Studies Discipline, Social Science School, Khulna University, Khulna, Bangladesh

\* mortuzacreations@gmail.com

**Data Availability Statement:** This study utilized the publicly available Demographic and Health Surveys (DHS) Program dataset of Bangladesh, which can be obtained freely from https://dhsprogram.com/.

## Abstract

Maternal and child mortality rates remain a significant concern in South Asian countries, primarily due to limited access to maternal care services and socioeconomic disparities. While previous studies have examined the factors influencing the utilization of antenatal care (ANC) services in individual countries, there is a lack of comparative analysis across South Asian nations. This study aims to investigate the factors affecting ANC utilization among women aged 15–49 in Bangladesh, India, Nepal, Maldives, and Pakistan using the latest Demographic and Health Survey data. The study utilized a total weighted sample size of 262,531 women. Simple bivariate statistics and binary logistic regression were employed to identify potential factors influencing ANC utilization. Decomposition analysis and concentration curve (Lorenz curve) were used to assess inequality in ANC service utilization. The prevalence of ANC utilization varied across the countries, with Maldives having the highest (96.83%) and Bangladesh the lowest (47.01%). Women's and husbands' education, household wealth status, BMI, and urban residence were found to significantly influence maternal healthcare services utilization. Higher education levels, affluent wealth quintiles, and urban living were identified as significant contributors to socioeconomic disparities in accessing ANC services. This study highlights the crucial role of socioeconomic factors in the utilization of maternal healthcare services in South Asian countries. Governments should focus on improving healthcare infrastructure, addressing cultural barriers, and promoting education to address these disparities. Identifying context-specific causes of maternal healthcare utilization is essential to inform targeted interventions and policies aimed at improving access to ANC services and reducing maternal mortality rates.

## Introduction

Maternal mortality is a significant problem worldwide, with over 800 deaths occurring daily. The maternal mortality rate (MMR) has declined by 38% globally since 2000, but in 2017, 295,000 women died due to complications during childbirth and pregnancy [1]. The majority

As the third-party user, we do not have permission to share the data publicly on any platform. Data are accessible free of charge upon a registration with the Demographic and Health Survey program (The DHS Program).

**Funding:** The author(s) received no specific funding for this work.

**Competing interests:** The authors have declared that no competing interests exist.

of these deaths took place in lower and middle-income nations, especially in Southern Asia and Africa. Inadequate maternal health services, comprising antenatal care (ANC), skilled birth attendance (SBA), facility delivery and postnatal care (PNC), are contributing factors [2–4].

The Sustainable Development Goals (SDGs) target to bring down MMR to at elast 70 for every 100,000 live births worldwide by 2030 [5]. Three South Asian countries, Afghanistan, India, and Pakistan, face unique challenges in improving maternal health outcomes. While progress has been made, these countries continue to experience significant disparities in maternal health care utilization between women of different socioeconomic backgrounds.

India has made significant progress in reducing maternal mortality since the 1990s but still has a high MMR rate of 130 per 100,000 live births in 2016 [6]. Pakistan has an even higher MMR rate of 276 per 100,000 live births [7], while Afghanistan has the worst position with a 359 MMR rate [8]. However, several other Asian nations, such as Bangladesh, Nepal, and Sri Lanka, have successfully reduced their MMR, meeting the requirement for the Millennium Development Goals (MDGs) [2]. Improvements in maternal health care services, such as institutional delivery, lower costs of skilled birth attendance, and investment in critical factors that enhance women's decision-making, have contributed to the success of these countries in reducing maternal mortality rates [9]. Despite the challenges, there is hope that with continued efforts, maternal mortality rates in South Asia and other regions can continue to decline towards the SDG targets.

Maternal death is a significant public health concern, and studies have shown that there are disparities in access to and utilization of mother care services [10–12]. Adequate antenatal care (ANC) services, including at least four visits throughout pregnancy, delivery in a healthcare center with skilled birth attendance, and proper postnatal care, can significantly reduce the risk of maternal death [10,13].

Evaluating the factors that influence ANC utilization at the individual and community levels is crucial, and this research performed in different regions and contexts since socioeconomic variables may have different effects [14,15]. Studies conducted in South Asia suggest that poverty, limited access to healthcare facilities, and lack of education and information are major drivers of inequalities in ANC service utilization [16,17]. Women from lower socioeconomic backgrounds are less likely to seek out ANC care services and are also more likely to receive substandard care.

Cultural attitudes and beliefs, including traditional practices and patriarchal values, also play a significant role in shaping maternal health behaviors and outcomes [18,19]. In Ethiopia and Turkey, women's perceptions towards ANC, particularly the direct benefits they can receive, and the acute need for these services, play crucial roles [20]. Household and self-income, the average distance to the nearest healthcare center, economic status, and household members are significant variables that determine ANC seeking behavior.

Efforts to reduce disparities in maternal healthcare utilization in South Asia have focused on improving access to maternal healthcare services, increasing awareness and education on maternal health issues, and promoting gender-sensitive policies and programs [21]. However, socioeconomic inequalities in antenatal care (ANC) utilization remain a significant challenge in the region, particularly for women from lower socioeconomic backgrounds. The five chosen countries, namely Bangladesh, India, Pakistan, Maldives, and Nepal, are confronted with substantial socioeconomic challenges encompassing poverty, escalating inequalities, crime rates, and disparities in education. Given their shared geographical location, it becomes imperative to discern and compare the impact of these correlated factors on the utilization of Antenatal Care (ANC) services.

Addressing these disparities will require a multi-faceted approach that leverages the strengths of the health system and the community, and addresses the underlying social and

cultural determinants of health. Therefore, this study aims to ascertain the determinants of Antenatal care utilization among reproductive-aged women in five South Asian countries, using the most recent database. The study considers at least four ANC visits as the outcome variable and takes into account all the influencing factors to provide a comprehensive understanding of the factors that affect ANC utilization. The results of this study can help guide the creation of laws and other measures to guarantee that all women in the area have access to ANC services.

## Methods

### Sources of data and sampling technique

Secondary data were mainly used in this study and obtained from Demographic and Health Surveys (DHS) conducted in Bangladesh, Maldives, India, Nepal, and Pakistan. The sources used were BDHS 2017–18, IDHS 2019–21, MDHS 2016–17, NDHS 2018, and PDHS 2017–18, which are the most up-to-date secondary databases available in these countries.

The data collection process was carried out under strict supervision and in compliance with all applicable laws. The DHS surveys used a multi-stage, probabilistic sampling approach to collect in-depth data. The first stage involved dividing each country into 18 distinct regions, which included both urban and rural areas. These clusters served as the foundation of the sample and were selected with a probability that reflected their relative importance to the entire population. In the second stage of sampling, a complete list of all families in each cluster was compiled, and 25 families were randomly selected with a probability of 19%. More detailed information about the sampling and data collection can be found in the relevant DHS report for each nation.

### Study area profile

This study encompasses five distinct countries situated in South Asia: Bangladesh, India, Nepal, Maldives, and Pakistan. Despite their shared geographical location and features, these nations exhibit diverse socioeconomic characteristics. In terms of poverty, inequality, and education, these South Asian nations of exhibit varying degrees of development. India, the largest and most economically advanced, struggles with significant poverty and income inequality, alongside a wide education gap. Bangladesh has made strides in poverty reduction and education, but still faces challenges in bridging inequality gaps. Nepal contends with deep-rooted poverty and limited educational access, striving to improve conditions. The Maldives, with a focus on tourism, has seen progress in poverty alleviation and education, though regional disparities persist. Pakistan confronts pervasive poverty, unequal educational opportunities, and social inequality, impeding its overall socioeconomic progress. These countries exemplify a diverse range of circumstances, reflecting their unique historical, political, and economic contexts.

### Dependent variable

The study focuses on ANC service utilization as the dependent variable. WHO suggests a minimum of four ANC checkups during pregnancy, starting from the 12th to the 40th week. In recent addition they have suggested at least 8 visits but we are considering four as the standard [22]. Therefore, if a woman visits four time for availing ANC services during her pregnancy, then that will be considered as appropriate otherwise not.

## Independent variable

Systematic literature review carried out through two selected electronic databases (PubMed and Scopus) to identify the most influential socioeconomic variable for ANC use. Distinct variables were suggested by different articles but researcher holds onto a saturation point for nominating one as independent variable. After thorough review of literature residential place, age, education, body mass index, occupation, husbands' education and occupation and household wealth status finally meet all the criteria and selected as explanatory variable. The construct of those explanatory variable is depicted in the **S1 Table**.

## Statistical analysis

Raw data on maternal health care utilization was filtered by restricting the age range of mothers to between 15 and 49 years and removing any missing or unreported data for relevant variables. A total of 262,531 data (unweighted) were analyzed using sample weight, clustering, and stratification data from DHS to ensure national representativeness. Primary sampling units were clustered to account for interdependence of error terms within households. Unobservable country-level covariates were considered through reweighting observations based on population size and including country-fixed effects. Binary logistic regression models were used for pooled analysis, including both single-adjusted and fully adjusted models. Descriptive statistics were used to analyze data characteristics and the distribution of ANC utilization. The Lorenz curve was used to determine inequality in maternal health care utilization based on household wealth status. Deviation from the 45° line on the curve indicated the existence of inequality, with the concentration curve being below the line indicating higher utilization among respondents from higher wealth quartile and above the line indicating lower utilization among those with lower wealth status.

The level of concentration was measured using the CIX index, which involved using the convenient covariance method to address the CIX. The formula used for this is presented below.

$$CIX = \frac{2}{\mu} cov(h, r);$$

Here, $\mu$ = Weighted mean of the health care utilization,
$h$ = Health care utilization variables,
$r$ = Fractional rank of the individual in the distribution of household wealth status,
cov(h, r) = covariance between h and r.

The study utilized STATA commands, namely Lorenz23 and conindex24, to generate the Lorenz curve and determine the level of concentration using CIX. The CIX value ranges from -1 to +1, where a CIX value near to +1 indetifies a greater concentration in the upper quantile of the variable used for measuring concentration, while a CIX value closer to -1 indetifies a greater concentration in the lower quantile.

## Ethics approval and consent to participate

This study made use of secondary data obtained from the Demographic Health Survey (DHS) platform, which is freely accessible upon request. Since the data in this platform has already received ethical approval from the relevant countries and has been deidentified, thus, we do not need to seek additional ethical approval.

## Results

Table 1 presents descriptive statistics on the characteristics of women utilizing maternal health care in five South Asian nations. The data set comprises 262,531 observations. The largest proportion of women (88.72%) in the sample were from India. Of these, the majority resided in rural areas (77.52%). In terms of socioeconomic status, most of the women belonged to the poorest (26.84%) and poorer (23.26%) groups. 25–34 years age group highest number ofwomen(58.52%). The majority of women in the sample had a normal BMI (62.72%).

**Table 1. Background characteristics of study participants.**

| Characteristics | | Number | % |
|---|---|---|---|
| **Country** | | | |
| | Bangladesh | 8,759 | 3.34 |
| | India | 232,920 | 88.72 |
| | Maldives | 3,106 | 1.18 |
| | Nepal | 5,038 | 1.92 |
| | Pakistan | 12,708 | 4.84 |
| **Place of Residence** | | | |
| | Urban | 59,015 | 22.48 |
| | Rural | 203,516 | 77.52 |
| **Maternal Age** | | | |
| | 15–24 | 81,546 | 31.06 |
| | 25–34 | 153,642 | 58.52 |
| | 35–49 | 27,343 | 10.42 |
| **Body Mass Index** | | | |
| | <18.50 (Underweight) | 44,688 | 18.21 |
| | 18.50–24.90 (Normal) | 153,949 | 62.72 |
| | 25.00–29.99 (Overweight) | 36,172 | 14.74 |
| | <30 (Obesity) | 10,626 | 4.33 |
| **Women's Highest Education Level** | | | |
| | No education | 60,107 | 22.90 |
| | Primary | 35,941 | 13.69 |
| | Secondary | 130,223 | 49.60 |
| | Higher | 36,260 | 13.81 |
| **Current Working Status** | | | |
| | Not working | 49,616 | 76.06 |
| | Working | 15,615 | 23.94 |
| **Husband's Education Level** | | | |
| | No education | 11,399 | 17.61 |
| | Primary | 11,310 | 17.48 |
| | Secondary | 30,762 | 47.54 |
| | Higher | 11,242 | 17.37 |
| **Occupation of the Husband** | | | |
| | Agricultural | 11,604 | 18.25 |
| | Non-Agricultural | 51,971 | 81.75 |
| **Wealth Status** | | | |
| | Poorest | 70,469 | 26.84 |
| | Poorer | 61,072 | 23.26 |
| | Middle | 51,097 | 19.46 |
| | Richer | 44,321 | 16.88 |
| | Richest | 35,572 | 13.55 |

In terms of education, almost half of the women (49.60%) had a secondary level of education. However, only a small proportion of women were employed (23.94%). Regarding the education level of partners, the most prevalent level was secondary education (47.54%). The majority of partners were involved in non-agricultural occupations (81.75%).

## Contributing factors to ANC service utilization: Pooled data

Maternal healthcare utilization was found to be influenced by various demographic as well as socio-economic variables. Table 2 presents women living in urban areas have 1.22 times (CI: 1.15–1.29) better odds to receive antenatal care (ANC) compared to those living in rural regions, while women in the maternal age group of 25–34 had 1.09 times (CI: 1.01–1.17) higher probability of having ANC services compared to those aged 35–49.

Obese and overweight women were also observed to have enhanced odds of getting ANC, with odds ratios of 1.18 (CI: 1.11–1.26) and 1.25 (CI: 1.12–1.39), respectively (Table 2). Education level of both the wife and the husband significantly influence ANC utilization, as did the wealth status of the household. There was a high correlation between wealth status and ANC service utilization. Working women had 1.17 (CI: 1.10–1.23) times better chance of receiving antenatal care compared to unemployed women, and women whose husbands worked outside the home had 1.12 times (CI: 1.06–1.19) higher odds of having ANC.

## Predictors of ANC service use in the countries investigated

In Bangladesh (S2 Table), urban women have better odds of using ANC services offered to them compared to rural women. Women with normal BMI have less possibility to use ANC services compared to women who were overweight or obese. Education had a positive influence on the utilization of ANC services, and women who worked in any capacity have better odds to use ANC services. Women whose husbands/partners had more education than those who had none have much higehr probability to get ANC visits compared to women whose partners had no education. ANC visits were strongly related to household affluence, with women from most affluent homes using ANC services more frequently than women from poor families.

In India (S3 Table), Women residing in urban regions exhibited a higher probability of obtaining antenatal care (ANC) checkups compared to their rural counterparts. Women who were overweight or obese had higher odds of using ANC than did women with a normal BMI. Better-educated women and their husbands use ANC checks more frequently than less-educated women and their husbands do. The utilization of mother care services was strongly correlated with wealth level.

Nepal (S4 Table) shows that urban women have almost two-time better odds to have antenatal care (ANC) visits than rural women (OR: 1.88, CI: 1.55–2.27). Additionally, this research indicated that women with greater educational attainment exhibited a higher tendency to utilize ANC services, and those from wealthier families were also more likely to utilize ANC.

The data from Pakistan (S5 Table) indicates that urban women had 1.47 times higher odds of having ANC visits compared to rural women (CI: 1.22–1.78). The study also found that young aged women have more probability to use ANC services, and that overweight and obese women have 1.24 (CI: 1.01–1.52) times higher odds to have proper ANC visits compared to women with a normal body mass index. The utilization of ANC services in Pakistan was strongly correlated with the education level of both the wife and husband, as well as the wealth of the household.

Fig 1 depicts inequalities of using ANC services using a Lorenz curve or concentration curve (CC). The results showed that in four of the studied countries, the Lorenz curve was

**Table 2. Factors associated with ANC utilization: Pooled analysis.**

| Characteristics | | AOR ANC (95% CI) |
|---|---|---|
| **Place of Residence** | | |
| | Urban | 1.22 (1.15–1.29) *** |
| | Rural (RC) | |
| **Maternal Age** | | |
| | 15–24 | 1.07 (0.98–1.15) |
| | 25–34 | 1.09 (1.01–1.17)* |
| | 35–49 (RC) | |
| **Body Mass Index** | | |
| | <18.50 (Underweight) | 0.98 (0.93–1.04) |
| | 18.50–24.90 (Normal) (RC) | |
| | 25.00–29.99 (Overweight) | 1.18 (1.11–1.26)*** |
| | <30 (Obesity) | 1.25 (1.12–1.39)*** |
| **Women's Educational Attainment** | | |
| | No education (RC) | |
| | Primary | 1.31 (1.21–1.41) *** |
| | Secondary | 1.68 (1.57–1.80) *** |
| | Higher | 2.02 (1.84–2.22) *** |
| **Current Working condition** | | |
| | Not working (RC) | |
| | Working | 1.17 (1.10–1.23) *** |
| **Education of husband** | | |
| | No education (RC) | |
| | Primary | 1.15 (1.07–1.25) *** |
| | Secondary | 1.31 (1.22–1.41) *** |
| | Higher | 1.33 (1.21–1.46) *** |
| **Occupation of the Husband** | | |
| | Agricultural (RC) | |
| | Non-Agricultural | 1.12 (1.06–1.19) *** |
| **Wealth Distribution** | | |
| | Poorest (RC) | |
| | Poorer | 1.25 (1.18–1.34) *** |
| | Middle | 1.70 (1.59–1.82) *** |
| | Richer | 1.93 (1.78–2.08) *** |
| | Richest | 2.54 (2.31–2.79) *** |

*p<0.05
**p<0.01
***p<0.001.

below the line of equality, indicating that women from wealthier households have better odds to seek ANC. However, the overall trend was positive, with the curve heading towards equality. The largest gap between the Lorenz curve and the equality line was found in Pakistan, while the smallest gap was found in the Maldives.

## Decomposition analysis

Table 3 provides information on how major socioeconomic and demographic factors affect maternal health care utilization and inequalities. The "Elasticity" column shows how a one-

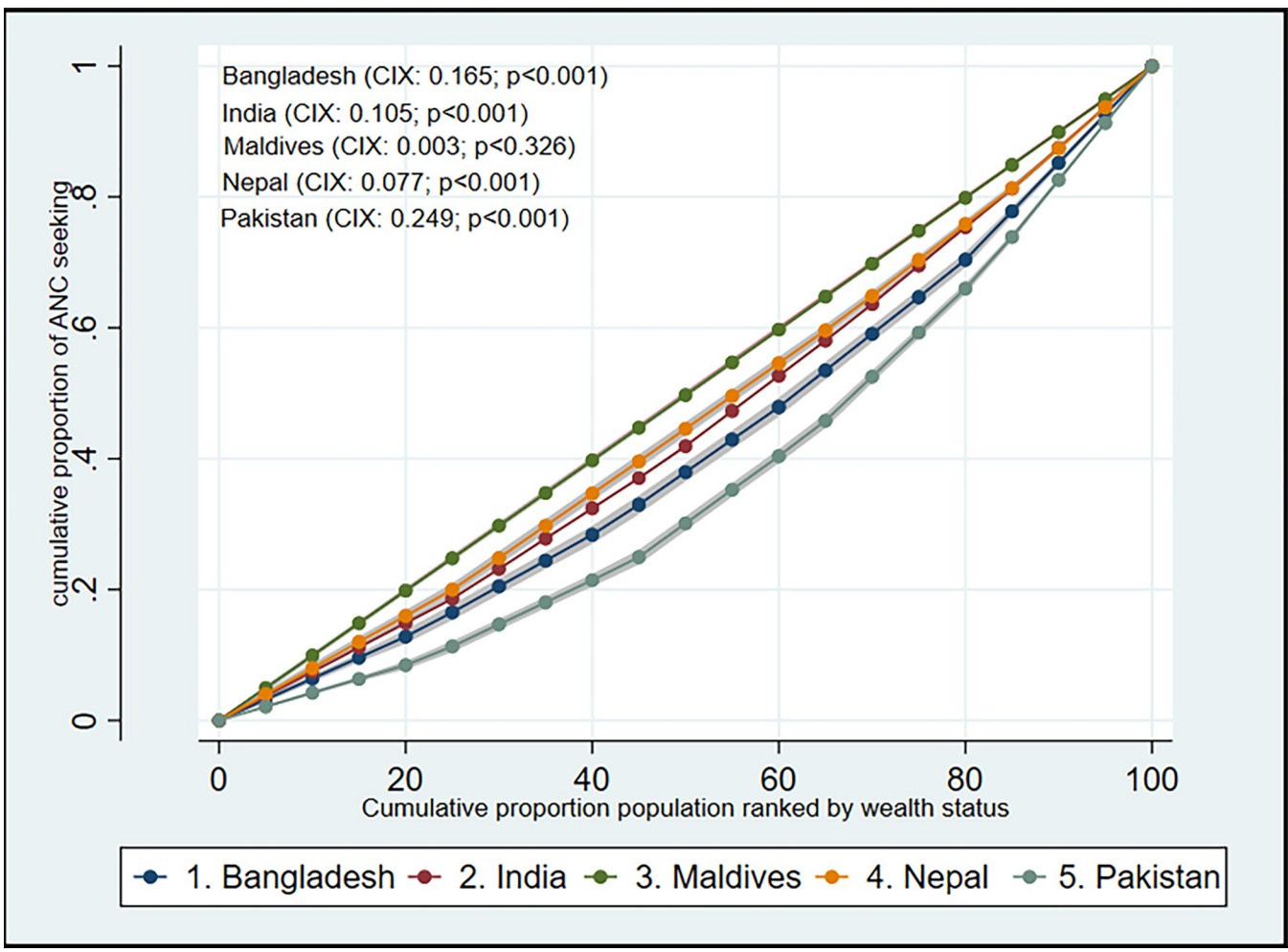

**Fig 1. Concentration curve for inequality estimation of ANC seeking behaviour.**

unit shift in the exploratory factors affects the dependent variable, which is the socioeconomic imbalance in ANC service utilization. The elasticity column displays a positive or negative value, indicating whether a factor is positively or negatively correlated with the facility's output trend.

The CIX values range from -1 to +1 and measure the level of socioeconomic inequity. A CIX value of 0 indicates equal distribution of facilities among socioeconomic groups. In this study, the CIX value for ANC service utilization was found to be 0.2346 (p<0.001) among women with higher socioeconomic status, indicating a socioeconomic inequality in favor of the more affluent. The column labeled "CIX" illustrates how the determinants are distributed across different wealth quintiles. The direction of the CIX indicates whether the factors are more prevalent in the wealthier or poorer group.

The column titled "Percentage contribution" specifies the extent to which each variable in the model contributes to the socioeconomic disparities observed in the utilization of maternal healthcare. A positive percentage contribution suggests that the variable widens socioeconomic gaps in ANC usage, while a negative percentage contribution suggests a reduction in socioeconomic disparities.

**Table 3. Decomposition analysis of ANC seeking behaviour: Pooled analysis.**

| | Elasticity | CIX | Contribution to overall CIX = 0.2346 (p<0.001) | |
| --- | --- | --- | --- | --- |
| | | | Absolute contribution | Percentage contribution |
| **Characteristics** | | | | |
| **Name of the Country** | | | | |
| Bangladesh | -0.0123 | 0.0609 | -0.0007 | -0.6982 |
| India | 0.1548 | -0.0050 | -0.0008 | -0.7149 |
| Maldives | 0.0286 | 0.0459 | 0.0013 | 1.2232 |
| Nepal | 0.0111 | 0.0262 | 0.0003 | 0.2708 |
| Pakistan (RC) | | | | |
| *Sub-total* | | | | *0.0809* |
| **Place of Residence** | | | | |
| Urban | 0.0521 | 0.4316 | 0.0225 | 20.9906 |
| Rural (RC) | | | | |
| *Sub-total* | | | | *20.9906* |
| **Maternal Age** | | | | |
| 15–24 | 0.0076 | -0.0721 | -0.0005 | -0.5095 |
| 25–34 | 0.0094 | 0.0449 | 0.0004 | 0.3954 |
| 35–49 (RC) | | | | |
| **Body Mass Index** | | | | |
| <18.50 (Underweight) | -0.0145 | -0.2026 | 0.0029 | 2.7322 |
| 18.50–24.90 (Normal) (RC) | | | | |
| 25.00–29.99 (Overweight) | 0.0268 | 0.2586 | 0.0069 | 6.4646 |
| <30 (Obesity) | 0.0101 | 0.3761 | 0.0038 | 3.5425 |
| *Sub-total* | | | | 12.7393 |
| **Women's Educational Attainment** | | | | |
| No education (RC) | | | | |
| Primary | 0.0423 | -0.2181 | -0.0092 | -8.6001 |
| Secondary | 0.2739 | 0.0706 | 0.0193 | 18.0315 |
| Higher | 0.1140 | 0.5099 | 0.0581 | 54.2215 |
| *Sub-total* | | | | *63.6529* |
| **Current Working Condition** | | | | |
| Not working (RC) | | | | |
| Working | 0.0303 | -0.0988 | -0.0030 | -2.7924 |
| **Education of husband** | | | | |
| No education (RC) | | | | |
| Primary | 0.0393 | -0.2143 | -0.0084 | -7.8588 |
| Secondary | 0.1321 | 0.0675 | 0.0089 | 8.3139 |
| Higher | 0.0461 | 0.4373 | 0.0202 | 18.7988 |
| *Sub-total* | | | | *19.2539* |
| **Occupation of the Husband** | | | | |
| Agricultural (RC) | | | | |
| Non-Agricultural | 0.0653 | 0.0053 | 0.0003 | 0.3200 |
| **Wealth Distribution** | | | | |
| Poorest (RC) | | | | |
| Poorer | 0.0485 | -0.2980 | -0.0144 | -13.4715 |
| Middle | 0.0898 | 0.1144 | 0.0103 | 9.5828 |
| Richer | 0.0912 | 0.4961 | 0.0453 | 42.2116 |
| Richest | 0.1204 | 0.8407 | 0.1012 | 94.3946 |
| *Sub-total* | | | | *132.7175* |

## Discussion

South Asian nations have diverse maternal healthcare systems, and their healthcare utilization patterns vary significantly. Despite extensive campaigns and initiatives to improve maternal health care in several South Asian countries, there is still reluctance among some to use these services. However, there has been significant progress in some countries. A study conducted in Bangladesh, India, Nepal, and Pakistan revealed that there has been an improvement in the utilization of ANC facilities over the last decade [23]. Nevertheless, to attain the sustainable development goal of bringing MMR to 70 per 100,000 live births by 2030, nations like India and Pakistan must enhance the quality and availability of maternal healthcare services. This will aid in planning, developing, and implementing necessary interventions and removing obstacles to improving health care utilization and decreasing maternal mortality rates.

The current study sheds light on various factors that are believed to influence the use of ANC services. As reported in many previous studies [17,24–28], this study also concluded, women residing in urban areas in all five observed nations have higher odds of using ANC services than rural women. This trend can be attributed to several reasons, such as the availability and capacity of healthcare centers in urban areas, higher levels of education and exposure to media among urban women [26,29], and a larger number of private and government healthcare facilities in cities [3,17,30,31].

Consistently, prior research has arrived at similar conclusion, for instance, a study conducted in rural Africa, discovered that women who resided in urban regions exhibited a higher likelihood of utilizing ANC services in comparison to those living in rural areas [32]. Another study conducted in Tanzania presented that, women residing in urban areas had better access to ANC facilities due to the availability of well-equipped healthcare facilities and trained healthcare workers [33]. One effective approach could be to increase the number of healthcare facilities in rural areas and to ensure that these facilities are well-equipped and staffed with trained healthcare providers [34]. Additionally, community-based interventions, such as health education campaigns and outreach programs, can be implemented to increase familiarity regarding the significance of ANC among women residing in rural areas.

The study has demonstrated the importance of both women's and husband's educational attainment in the utilization of Antenatal Care (ANC) services across all the countries investigated. However, it is noteworthy that women's education has a more significant impact on their decision to seek ANC services compared to their husband's education [17,29–31,35]. Possible reasons can be Educated women have greater knowledge of the importance of ANC for a healthy pregnancy and childbirth, better access to information and healthcare resources, and greater decision-making power to prioritize their health and their unborn child's health. Additionally, education plays a crucial role in improving maternal and child health outcomes by increasing awareness, access, and agency to seek necessary healthcare services.

Moreover, educated husbands can also play a significant role in motivating their wives to seek ANC services. They can communicate with their partners more effectively about maternal healthcare and help increase the value of these services. Previous Studies conducted in Ghana [36]. and Ethiopia [37] found that women with husbands who possessed higher levels of education demonstrated a greater propensity to utilize Antenatal Care (ANC) services. Husbands with higher education were more aware of the benefits of ANC and were supportive of their wives seeking medical care, leading to improved access to healthcare resources and higher utilization of ANC services by pregnant women. Therefore, the study recommends that both men and women receive education to increase access and utilization of maternal healthcare facilities.

In line with previous studies conducted in South Asian countries, this study has consistently shown a strong correlation between household wealth status and the use of ANC services [38].

Wealthier families have more resources to spend on healthcare services and may be able to afford higher-quality ANC services, such as more frequent visits, advanced diagnostic tests, and better medical care. Wealthier families may have greater access to healthcare providers, who can provide better health education and counseling on the significance of ANC utilization during pregnancy [39]. Additionally, wealthier families are more likely to choose better hospital facilities and have greater access to health care information, which can result in better utilization of ANC services during pregnancy.

Another finding this study present is women who work, have a higher likelihood of seeking proper antenatal care (ANC) services in all five countries studied. This finding is consistent with previous studies that suggest that women's autonomy and decision-making capacity are crucial factors in maternal healthcare-seeking behavior. The reason why working women may seek more ANC services could be due to their increased access to resources, such as income and education, which enable them to make informed decisions about their healthcare. Additionally, working women may have more opportunities to interact with healthcare providers and receive information about the importance of ANC services. Furthermore, working women may have greater control over their own health and pregnancy outcomes due to their financial independence and ability to take time off from work for healthcare appointments.

Young women aged between 15–24 are more likely to seek antenatal care (ANC) services in India and Pakistan than women of older ages. However, maternal age does not seem to be a crucial factor in determining the utilization of ANC facilities in Bangladesh and Nepal.

In India [40] and Pakistan [41] studies have presented that younger women are more inclined to obtain antenatal care due to early marriage and pregnancy, inadequate education and awareness, lower socio-economic status, poor knowledge about maternal health, and cultural beliefs that prioritize the health of the fetus over the mother's health. Moreover, early pregnancy is associated with increased susceptibility to viral and parasitic infections and a higher likelihood of experiencing pregnancy-related complications [3,42]. On the other hand, the absence of hemoglobinopathies that cause anemia has been shown to be a motivating factor for women to seek maternal care services.

In contrast, studies conducted in Bangladesh and Nepal have shown that maternal age is not an important factor in indicating the use of ANC facilities. Instead, factors such as socioeconomic status, educational attainment, and access are more influential in determining the utilization of ANC services [43,44].

This study found a significant link between a woman's body mass index (BMI) and her use of antenatal care (ANC) services in four different countries. Regardless of the country, the utilization of maternal healthcare services was more frequent among women with higher BMI in comparison to those with a normal BMI. Several early studies also acknowledged this correlation [45–47]. One possible reason can be the increased risk of complications during pregnancy and childbirth associated with obesity [48,49]. Research has indicated that obesity is linked to an elevated risk of gestational diabetes, postpartum hemorrhage, cesarean delivery, and preeclampsia [50,51]. Therefore, healthcare providers may recommend and encourage women with high BMI to attend ANC services to monitor their health and the health of their baby throughout pregnancy. Additionally, ANC visits may provide opportunities for healthcare providers to discuss and educate women on lifestyle changes that can reduce the risk of complications during pregnancy and childbirth [52].

Overall, regional disparities in ANC services utilization exist due to factors such as socioeconomic status, education, location, and access to healthcare resources. The study's primary strength was the utilization of the latest demographic data available on a nationwide scale, allowing for broad generalizations of the results due to the large sample size. However, the

survey was cross-sectional, which means that no causal relationships can be inferred as both the predictors/exposures and outcomes were measured simultaneously.

## Policy recommendation

This study examined ANC utilization in South Asia and found that it is low, especially among marginalized women. The study offers original findings and confirms previous research. The study highlights the need for a comprehensive strategy to increase ANC utilization, including improving access, awareness, and addressing socio-economic barriers. Therefore, this study recommends the following policy measures:

- Strengthening the healthcare infrastructure: The governments of these countries should invest in improving the healthcare infrastructure in rural areas, including building and upgrading health facilities and training healthcare workers. This will help to increase the availability and accessibility of ANC services, particularly in areas where there are currently no facilities.

- Providing financial incentives: Financial incentives such as cash transfers or vouchers can encourage women to seek ANC services, particularly those who are poor. Governments can provide incentives to women who attend a certain number of ANC visits or who give birth in a health facility.

- Addressing cultural and social norms: Cultural and social norms can discourage women from seeking ANC services, particularly in conservative societies. Governments should work with communities to enhance knowledge and sensation about the significance of ANC and address any cultural or social barriers that may prevent women from seeking care.

- Increasing education: Education can improve health-seeking behavior and increase knowledge about the importance of ANC services. Governments should invest in education, particularly for girls and women, to promote better health-seeking behavior and increase ANC service uptake.

## Conclusion

This study highlights significant socio-economic barriers that prevent reproductive-aged women from seeking adequate antenatal care (ANC) services in five countries—India, Pakistan, Bangladesh, Maldives, and Nepal. The study found that factors such as residential place, women's and partner's education level, BMI, household wealth status, age, and working conditions greatly influenced the utilization of ANC services in all the countries. Although younger women in India and Pakistan sought more ANC services than older women, age did not significantly impact ANC utilization in Bangladesh and Nepal.

The study revealed that the overall utilization of ANC services was particularly poor in India, Bangladesh and Pakistan compared to Maldives and Nepal. However, all five countries must focus on comprehensive policy implementation and fund distribution to improve the situation. It is essential to create an enabling environment that ensures equal access to ANC services for all women, although this presents significant challenges. To ensure proper maternal health, the government must achieve a perfect or at least an enabling environment that addresses the specific causes of variable maternal healthcare utilization. Further research is necessary to explore the context-specific causes of maternal healthcare utilization to make safe motherhood a reality in developing countries.

## Supporting information

**S1 Table. Construct of the independent variables.**
(DOCX)

**S2 Table. Factors associated with ANC: Bangladesh.**
(DOCX)

**S3 Table. Factors associated with ANC: India.**
(DOCX)

**S4 Table. Factors associated with ANC: Nepal.**
(DOCX)

**S5 Table. Factors associated with ANC: Pakistan.**
(DOCX)

## Author Contributions

**Conceptualization:** Mortuja Mahamud Tohan, Md. Amirul Islam.

**Data curation:** Mortuja Mahamud Tohan.

**Formal analysis:** Mortuja Mahamud Tohan.

**Methodology:** Mortuja Mahamud Tohan.

**Supervision:** Md. Ashfikur Rahman.

**Validation:** Md. Ashfikur Rahman.

**Writing – original draft:** Mortuja Mahamud Tohan, Md. Amirul Islam.

**Writing – review & editing:** Md. Ashfikur Rahman.

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
