## [Decision Letter · Decision Letter 0]

16 Aug 2023

PONE-D-23-19791Exploring the Factors Behind Socioeconomic Disparities in Antenatal Care (ANC) Utilization across Five South Asian Nations: A Decomposition Approach.PLOS ONE

Dear Dr. tohan,

Thank you for submitting your manuscript to PLOS ONE. After careful consideration, we feel that it has merit but does not fully meet PLOS ONE’s publication criteria as it currently stands. Therefore, we invite you to submit a revised version of the manuscript that addresses the points raised during the review process.

We look forward to receiving your revised manuscript.

Kind regards,

Pradip Chouhan

Academic Editor

PLOS ONE

“We thank the MEASURE DHS Data Archive, ICF International, for providing access to the South and Southeast Asian Demographic and Health Surveys data. We also wish to express our appreciation to Khulna University Research and Innovation Center for providing the financing necessary to carry out this study. However, funders are not required in any way to participate in the study's design”

5. Please upload a copy of Supporting Information Tables 1-5 which you refer to in your text on pages 5,9 and 10.

Reviewers' comments:

Reviewer's Responses to Questions

**Comments to the Author**

1. Is the manuscript technically sound, and do the data support the conclusions?

Reviewer #1: Yes

Reviewer #2: Partly

2. Has the statistical analysis been performed appropriately and rigorously? 

Reviewer #1: Yes

Reviewer #2: Yes

3. Have the authors made all data underlying the findings in their manuscript fully available?

Reviewer #1: Yes

Reviewer #2: Yes

4. Is the manuscript presented in an intelligible fashion and written in standard English?

Reviewer #1: Yes

Reviewer #2: Yes

5. Review Comments to the Author

Reviewer #1: A section mentioning the Socioeconomic statuus of each of the population group in the five countries would be helpful.It may be included in the methodology section(Profile of the Study area ) as it would set the context of the study.

Reviewer #2: This manuscript examines very serious issue of public health as the authors indicate that there is a great variation in the utilization of antenatal care services among the five South Asian Nations. The present work shows that due to variation in the different socio-demographic aspects, the variations in the utilization of antenatal care services also found in the different countries. The manuscript used recent round of nationally representative database to assess the determining factors of utilization of antenatal care services. In my assessment, this manuscript highlighted some important findings which will help for policy implementations. However, the authors are suggested to minor revision in their manuscript. Comments are mentioned below:

1. The authors were taken at least four ANC visits during pregnancy as antenatal care variable in the present study but why not taken full ANC i.e. at least four ANC visits during pregnancy, at least two tetanus injections and at least 100 or more IFA tablets among the different South Asian countries?

2. The authors are suggested to add the rationality for the selection of five South Asian nations in the present study.

3. Women’s autonomy, mass media exposure and male involvement play a vital role for the utilization of antenatal care services among the women but in the present study the authors did not included in their model but why?

4. Discussion section need to be improve in proper manner.

5. I would like to emphasize that the paper needs to check for some typo errors, proofreading, and minor explanations as per the above suggestions.

6. PLOS authors have the option to publish the peer review history of their article (what does this mean?). If published, this will include your full peer review and any attached files.

Reviewer #1: **Yes: **Parama Bannerji

Reviewer #2: No

---

## [Author Response · Author response to Decision Letter 0]

16 Sep 2023

5. Review Comments to the Author

Reviewer #1: A section mentioning the Socioeconomic status of each of the population group in the five countries would be helpful. It may be included in the methodology section (Profile of the Study area) as it would set the context of the study.

 Thank you for your comment. We have added an additional section including those background information of the selected countries.

Reviewer #2: This manuscript examines very serious issue of public health as the authors indicate that there is a great variation in the utilization of antenatal care services among the five South Asian Nations. The present work shows that due to variation in the different socio-demographic aspects, the variations in the utilization of antenatal care services also found in the different countries. The manuscript used recent round of nationally representative database to assess the determining factors of utilization of antenatal care services. In my assessment, this manuscript highlighted some important findings which will help for policy implementations. However, the authors are suggested to minor revision in their manuscript. Comments are mentioned below:

 Thank you for your comment and acknowledgment. 

1. The authors were taken at least four ANC visits during pregnancy as antenatal care variable in the present study but why not taken full ANC i.e. at least four ANC visits during pregnancy, at least two tetanus injections and at least 100 or more IFA tablets among the different South Asian countries?

 Thanks for your comment, we have categorized the ANC visit mothers who have taken at least four times ANC schooling accordingly by WHO definition. It is unclear to us or we didn’t find any definition that mother’s have to have at least two tetanus injections and at least 100 or more IFA tablets during pregnancy period. In addition, such information is not available in the DHS data set. 

2. The authors are suggested to add the rationality for the selection of five South Asian nations in the present study.

 Thank you for your comment we have added a section mentioning rationale for selecting those five countries in particular. (page 4)

3. Women’s autonomy, mass media exposure and male involvement play a vital role for the utilization of antenatal care services among the women but in the present study the authors did not included in their model but why?

 Thank you for your observation, however we have considered women’s working status as an indicator of their autonomy and we found it significant for predicting ANC utilization. However, in our analysis, we encountered limitations in the available DHS data, particularly concerning variables like media exposure, which were incomplete or unavailable, particularly in Pakistan and Nepal. Regrettably, these variables had to be omitted from our study. However, future research endeavors could potentially address and incorporate these issues, thereby enriching our understanding of the complex factors influencing the utilization of Antenatal Care (ANC) services across these countries.

4. Discussion section need to be improve in proper manner.

 We have tried to refine the discussion sections by incorporating robust supporting arguments and ensuring accurate referencing to existing knowledge.

5. I would like to emphasize that the paper needs to check for some typo errors, proofreading, and minor explanations as per the above suggestions.

 We have thoroughly proofread this manuscript, diligently addressing and rectifying any potential typographical errors.

---

## [Decision Letter · Decision Letter 1]

16 May 2024

Exploring the Factors Behind Socioeconomic Inequalities in Antenatal Care (ANC) Utilization across Five South Asian Nations: A Decomposition Approach.

PONE-D-23-19791R1

Dear Dr. tohan,

We’re pleased to inform you that your manuscript has been judged scientifically suitable for publication and will be formally accepted for publication once it meets all outstanding technical requirements.

Kind regards,

Jay Saha

Academic Editor

PLOS ONE

Additional Editor Comments (optional):

Reviewers' comments:

Reviewer's Responses to Questions

**Comments to the Author**

1. If the authors have adequately addressed your comments raised in a previous round of review and you feel that this manuscript is now acceptable for publication, you may indicate that here to bypass the “Comments to the Author” section, enter your conflict of interest statement in the “Confidential to Editor” section, and submit your "Accept" recommendation.

Reviewer #1: All comments have been addressed

2. Is the manuscript technically sound, and do the data support the conclusions?

Reviewer #1: Yes

3. Has the statistical analysis been performed appropriately and rigorously? 

Reviewer #1: Yes

4. Have the authors made all data underlying the findings in their manuscript fully available?

Reviewer #1: Yes

5. Is the manuscript presented in an intelligible fashion and written in standard English?

Reviewer #1: Yes

6. Review Comments to the Author

Reviewer #1: I have received satisfactory response from the author.

I hereby request ypu to accept the submission and publish it.

7. PLOS authors have the option to publish the peer review history of their article (what does this mean?). If published, this will include your full peer review and any attached files.

Reviewer #1: No

---

## [Editor Report · Acceptance letter]

26 Jun 2024

PONE-D-23-19791R1 

PLOS ONE

Dear Dr. tohan, 

I'm pleased to inform you that your manuscript has been deemed suitable for publication in PLOS ONE. Congratulations! Your manuscript is now being handed over to our production team.

Kind regards, 

on behalf of

Mr. Jay Saha 

Academic Editor

PLOS ONE